# Scalable Utility-Aware Multiclass Calibration

## Abstract

Ensuring that classifiers are well-calibrated, i.e., their predictions align with ob-
served frequencies, is a minimal and fundamental requirement for classifiers to
be viewed as trustworthy. Existing methods for assessing multiclass calibration
often focus on specific aspects associated with prediction (e.g., top-class confi-
dence, class-wise calibration) or utilize computationally challenging variational
formulations. We instead propose *utility calibration*, a general framework designed
to evaluate model calibration directly through the lens of downstream applications.
This approach measures the calibration error relative to a specific *utility function*
that encapsulates the goals or decision criteria relevant to the end user. As such,
utility calibration provides a task-specific perspective on reliability. We demon-
strate how this framework can *unify and re-interpret several existing calibration
metrics*, particularly allowing for more robust versions of the top-class and class-
wise calibration metrics, and to go beyond such binarized approaches, towards
assessing calibration for richer classes of downstream utilities.

## 1 Introduction

Calibration is a fundamental property of probabilistic predictors. A calibrated model produces
predictions that, on average, align with observed frequencies. For instance, if a weather forecaster
predicts a 30% chance of rain on a given day, rain should occur on approximately 30% of such days.
In multiclass classification problems, calibration ensures that the predicted probabilities reflect the
true likelihood of each class. Formally, let $\mathcal{X}$ denote the input space, $\mathcal{Y} = \{e_1, \ldots, e_C\}$ the output
space, where $e_i$ is the $i$-th canonical basis vector in $\mathbb{R}^C$, and $\Delta^{C-1} := \left\{ x \in \mathbb{R}^C_+ | \sum_i x_i \leq 1 \right\}$ denote
the simplex in $\mathbb{R}^C$. A predictor $f : \mathcal{X} \to \Delta^{C-1}$ is said to be perfectly calibrated with respect to
a distribution $D$ over $\mathcal{X} \times \mathcal{Y}$ if $\mathbb{E}[Y \mid f(X)] = f(X)$. The most direct metric for quantifying the
deviation from perfect calibration is the Mean Calibration Error (MCE).

**Definition 1.1** (Mean Calibration Error). *For a distribution $D$ such that $(X, Y) \sim D$ and a predic-
tor $f$, the mean calibration error is defined as* $\mathrm{MCE}(f) := \mathbb{E}\left[ \|\mathbb{E}[Y \mid f(X)] - f(X)\|_2 \right]$.

Without further assumptions, the MCE is fundamentally impossible to estimate, even in the binary
setting [1, 2]. While assumptions like Hölder continuity of $\mathbb{E}[Y|f(X)]$ allow for consistent estimators
of $\mathbb{E}[Y|f(X)]$ or minimax optimal tests for $\mathrm{MCE}(f)$ [1, 3, 4], their sample complexity scales
exponentially with the dimension $C$, making MCE estimation intractable in high dimensions.

Due to the difficulty of measuring MCE, multiple relaxations are proposed, falling into two main
categories: *binarized* and *variational*. First, binarized approaches [5–7] simplify the problem by
focusing on specific binary events derived from the multiclass predictions, e.g. top-class or class-wise
calibration. However, these methods are by nature presumptive of downstream tasks. Moreover, their
reliance on binning schemes or kernel estimators for the underlying binary subproblems introduce
sensitivity to estimator choices and can suffer from high bias [8]. Second, variational approaches
[9–14] assess calibration through optimization problems, such as the distance to the nearest perfectly

calibrated predictor or the worst-case error against a class of witness functions. Unfortunately, these methods can be computationally intensive and can scale poorly as the number of classes $C$ increases.

To address these limitations and provide an application-focused perspective on calibration, we introduce *utility calibration*. This framework evaluates a model $f$ by considering a downstream user who employs its predictions $f(X)$. The core idea is to measure calibration error relative to a specific *utility function*, denoted $u$, which encapsulates the goals, costs, or decision criteria relevant to this end user. Utility calibration then assesses how well the *expected utility* (as estimated by the user based on $f(X)$ and $u$) aligns with the *realized utility* (obtained when the true outcome $Y$ is observed). In practice, models often serve diverse users or a single user with multiple objectives. We thus extend utility calibration to handle *classes of utility functions*. The overall utility calibration for a class $\mathcal{U}$ can be defined as the worst-case error over $u \in \mathcal{U}$, denoted $\mathrm{UC}(f, \mathcal{U})$. A notable aspect of this class-based formulation is that it provides a structured way to express and analyze various existing calibration notions. In particular, by defining appropriate utility functions within $\mathcal{U}$, concepts such as top-class and class-wise calibration can be cast within the utility calibration framework. This offers a unified perspective and a superior alternative to binning for examining those notions of calibration.

**Contributions and outline:** In Section 2, we review related literature on calibration metrics and post-hoc calibration methods. In Section 3, we define utility calibration and relate it to existing measures of calibration. In addition, we demonstrate how this framework can be used to frame several existing calibration concepts within a common utility-centric perspective, offering consistent interpretations and providing examples of relevant utility classes. To characterize the difficulty of achieving utility calibration for classes of utility functions, we introduce the notions of *proactive* and *interactive* measurability. While, for rich utility classes, proactive measurability is not possible, we show that interactive measurability is achievable for many classes of interest. Drawing on these insights, we empirically demonstrate the application of our proposed metrics and evaluation methodology, in Section 4, to that end, we formulate a practical and scalable methodology for evaluating calibration against interactively measurable utility classes in Section 4.

**Notation:** For any vector $w \in \mathbb{R}^C$, $w_i$ denotes its $i$-th component and $\gamma(w) := \operatorname{argmax}_i w_i$. For a probability vector $p \in \Delta^{C-1}$, we write $Z \sim p$ to denote a categorical random variable $Z$ taking values in $\mathcal{Y} = \{e_1, \ldots, e_C\}$ such that $\mathbb{P}\{Z = e_i\} = p_i$, where $e_i$ is the $i$-th canonical basis vector. We use $\mathbb{1}\{\cdot\}$ for the indicator function. $\mathbb{E}[\cdot]$ denotes expectation, which is taken typically w.r.t. $(X, Y) \sim D$ and, for $k \in \mathbb{N}_+$, $[k] = \{1, \ldots, k\}$. Finally, for $a, b \in \mathbb{R}$ with $a < b$, we denote $\mathbb{I}[a, b]$ to be the set of closed interval subsets of $[a, b]$.

## 2 Related Work

In this section, we review three classical and related approaches to measuring or ensuring a form of calibration, namely binarized relaxations, variational approaches, and post-hoc calibration methods.

First, *binarized relaxations* aim to circumvent the difficulty of measuring the calibration error of a high-dimensional predictor $f$ by measuring the MCE of a single or multiple downstream binary versions of $f$ instead. Two commonly used relaxations are the Top-Class calibration Error (TCE) [7] and the Class-Wise calibration Error (CWE) [6], which are respectively defined as

$$\mathrm{TCE}(f) := \mathbb{E}\left[\left|\mathbb{E}[\mathbb{1}\{Y = e_{\gamma(f(X))}\} \mid f(X)_{\gamma(f(X))}] - f(X)_{\gamma(f(X))}\right|\right],$$

$$\mathrm{CWE}(f) := \sum_{i \in [C]} w_i \, \mathbb{E}\left[\left|\mathbb{E}[\mathbb{1}\{Y = e_i\} \mid f(X)_i] - f(X)_i\right|\right],$$

where $w_i$ is a class-dependent weight, which can be set to $1/C$, $w_i = \mathbb{P}\{Y = e_i\}$, or another choice. Typically, TCE and CWE are estimated using binning schemes. Concretely, for $(B_j)_{j \in [m]}$ $m$ disjoint subsets of $[0, 1]$ such that $\cup_{j \in [m]} B_j = [0, 1]$, we consider the following binned estimators

$$\mathrm{TCE}^{\mathrm{bin}}(f) = \sum_{j \in [m]} \left|\mathbb{E}\left[\left(f(X)_{\gamma(f(X))} - \mathbb{1}\{Y = e_{\gamma(f(X))}\}\right) \mathbb{1}\{f(X)_{\gamma(f(X))} \in B_j\}\right]\right|, \quad (2.1)$$

$$\mathrm{CWE}^{\mathrm{bin}}(f) = \sum_{i \in [C]} \sum_{j \in [m]} w_i \left|\mathbb{E}\left[\left(f(X)_i - \mathbb{1}\{Y = e_i\}\right) \mathbb{1}\{f(X)_i \in B_j\}\right]\right|. \quad (2.2)$$

Gupta and Ramdas [5] unified multiple instances of binarized proxies of MCE, such as TCE, CWE and $\mathrm{top}K$ confidence calibration, introduced in [15], and proposed additional binarized reductions

which offer stronger notions of calibration. Unfortunately, the binning schemes used in such binarized proxies are known to have a large effect on the estimated error [8, 16]. Apart from the simpler equal-size bins [7] and equal-weight bins [17], multiple binning schemes built on top of different heuristics have been proposed [see, e.g., 8, 18–20]. Gupta and Ramdas [21] showed a simple equal-weight binning scheme with better sample complexity guarantees for estimating bin averages. Kumar et al. [22] developed adaptive binning schemes with guarantees for discrete $f$ and showed that for any binning scheme, there exists a worst-case continuous $f$ such that the bias of $\mathrm{TCE}^{\mathrm{bin}}(f)$ as an estimate of $\mathrm{TCE}(f)$ is lower bounded by $0.49$ (noting that by construction TCE is bounded between $0$ and $1$). On the other hand, there exist binning-free alternatives for binarized reductions [see, e.g., 3, 15]. Nonetheless, in an assumption-free setting, it is generally impossible to consistently estimate the MCE of binary predictors [1, 2, 23]. As such, it is generally difficult to control the calibration error defined by binarized relaxations.

Second, *variational approaches* do not strictly aim to measure the MCE. Instead, they consider alternative formulations that do not require direct estimation of the conditional expectation. For example, Distance to Calibration (DC) quantifies the calibration error of a predictor $f$ as the distance between $f$ and the nearest perfectly calibrated predictors [10]:

$$\mathrm{DC}(f) := \inf_{\mathrm{MCE}(g)=0} \mathbb{E}\big[\|f(X) - g(X)\|_1\big].$$

A unified formulation of variational measures of calibration is weighted calibration, which assesses the calibration error against a class of witness functions [9]. Concretely, let $\mathcal{W}$ be a class of functions mapping $\Delta^{C-1}$ to $[-1, 1]^C$. Then, weighted calibration error with witness class $\mathcal{W}$ is

$$\mathrm{CE}_{\mathcal{W}}(f) \ = \ \sup_{w \in \mathcal{W}} \mathbb{E}_{X,Y}\left[\big\langle w\big(f(X)\big), f(X) - Y\big\rangle\right]. \tag{2.3}$$

A specific instance of weighted calibration is the Kernel Calibration Error (KCE) [24], which sets $\mathcal{W}$ to be the unit ball of the reproducing kernel Hilbert space (RKHS) of a multivariate universal kernel. This allows for efficient computation of the supremum but it remains hard to interpret the impact of low KCE for a user of $f$. Błasiok et al. [10] showed that in the binary setting, $\mathrm{DC}(f)$ and $\mathrm{CE}_{\mathrm{Lip}(1)}(f)$ are equivalent up to a (low-degree) polynomial scaling, where $\mathrm{Lip}(1)$ is the class of 1-Lipschitz functions from $\Delta^{C-1}$ to $[-1, 1]$. In addition, the authors proved that, for the binary setting, $\mathrm{CE}_{\mathrm{Lip}(1)}(f)$ can be well approximated by the RKHS of the Laplace kernel allowing for efficient assessment of $\mathrm{DC}(f)$ using a calibration metric originally proposed by Kumar et al. [12].

The result on the equivalence between $\mathrm{CE}_{\mathrm{Lip}(1)}(f)$ and $\mathrm{DC}(f)$ was further extended to the multiclass setting in [2, Theorem 15.5.5] and [11, Lemma 3.3]. In particular, Gopalan et al. [25] showed that measuring either $\mathrm{DC}(f)$ or $\mathrm{CE}_{\mathrm{Lip}(1)}(f)$ requires an exponential number of samples with respect to $C$ [11, Theorem 3.2. and Theorem 3.4.]. Thus, even though $\mathrm{DC}(f)$ can be efficiently assessed in the binary setting, it is quickly intractable as the dimension increases.

A particular case is *Decision calibration*, introduced by Zhao et al. [14], that tailors calibration guarantees to downstream decision-making tasks. A predictor $f$ is considered decision calibrated of order $K$ if, for any decision problem involving at most $K$ actions, the expected loss computed using the model's predictions $f(X)$ accurately matches the true expected loss incurred. Formally, for any loss function $\ell$ mapping an outcome-action pair to a real-valued loss, decision calibration of order $K$ requires:

$$\mathbb{E}\Big[\ell\big(\hat{Y}, \delta(f(X))\big)\Big] = \mathbb{E}\Big[\ell(Y, \delta(f(X)))\Big],$$

where $\hat{Y} \sim f(X)$ and $\delta$ is a decision rule that picks the best action among $K$ actions under the model's prediction $f(X)$. This ensures that decision-makers can reliably estimate the consequences of their choices when using the predictor. A key contribution of Zhao et al. [14] is showing that decision calibration of order $K$ can be achieved by having $\sup_{p \in P(K)} \|\mathbb{E}[(Y - f(X))\mathbb{1}\{f(X) \in p\}]\| = 0$, where $P(K)$ is the set of polytopes with at most $K$ supporting hyperplanes. Unfortunately, computational complexity is again an issue—Gopalan et al. [11] showed that even for $K = 2$ the computational complexity of measuring decision calibration is exponential with respect to $C$.

In summary, practitioners are faced with a dilemma in assessing the calibration error. On one hand, for binarized approaches, it is generally impossible to have consistent estimation of the calibration error of the binary subproblems. In addition, by preemptively only assessing specific binary subproblems, they are fundamentally presumptive of the downstream usage of the model. On the other hand,

variational approaches can offer more robust and well-motivated assessment of the calibration error but they are computationally infeasible as the dimension grows.

Independently, *post-hoc calibration* refers to techniques applied to a pre-trained model's outputs to improve the alignment between its predicted probabilities and the true likelihood of outcomes, without altering the original model parameters. Such methods are advantageous as they decouple the calibration process from the training process.

Common post-hoc calibration methods often adjust the model's outputs; popular examples include Temperature Scaling and its multi-parameter extensions, Vector Scaling and Matrix Scaling [7], which may all be regarded as a multiclass extension of Platt's scaling [26]. Dirichlet calibration assumes the model's predicted probability vectors can be modeled by a Dirichlet distribution, whose parameters are learned on a calibration set to transform the original probabilities [27]. Nonparametric methods such as Histogram Binning [17] and Isotonic Regression [28] learn calibration maps by discretizing the probability space or fitting monotonic (order-preserving) functions, respectively. Other methods also include: [18], which applies a specific binning strategy followed by recalibration to minimize class-wise calibration error, [29], which uses order-preserving transformations for recalibration to maintain accuracy. Finally, a related body of literature aims to improve calibration by changing or regularizing the training objective, e.g. [30, 3, 31, 12].

# 3 Utility Calibration

We consider the following utility-centric formulation of calibration. In particular, we are interested in the setting, where for some input $X$, a downstream user leverages $f(X)$ as an estimation of $\mathbb{E}[Y|X]$. Based on this estimation of the conditional expectation, the user may then take arbitrary actions or decisions. Finally, the user observes the true realization of the label $Y$ and based on this realization, may then suffer some loss or achieve some gain. To model such a pipeline of observation, action, then consequences, we consider a utility function $u : \Delta^{C-1} \times \mathcal{Y} \to [-1, 1]$ such that $u(f(X), Y)$ models the reward obtained or the loss suffered by the decision-makers after using $f(X)$ to take arbitrary actions/decisions. In such a setting, predictability is highly desirable, in the sense that when using the predictor $f$, the utility obtained is similar to the utility expected. More concretely, for $\hat{Y} \sim f(X)$ and a given input $X$, the user can use $f(X)$ to construct the following estimate of utility:

$$v_u(X) := \mathbb{E}\left[u(f(X), \hat{Y})|X\right] = \langle f(X), \vec{u}(X) \rangle, \tag{3.1}$$

where $\vec{u} : \mathcal{X} \to [-1, 1]^C$ is defined as $\vec{u}(X) := (u(f(X), e_i))_{i \in C}$. Ideally, we want the function $v_u(X)$ to be an unbiased estimator of the true utility. As such, we define the utility calibration with respect to a utility function $u$ as

$$\mathrm{UC}(f, u) := \sup_{I \in \mathbb{I}[-1,1]} |\mathbb{E}\left[(u(f(X), Y) - v_u(X))\mathbb{1}\{v_u(X) \in I\}\right]| \tag{3.2}$$

and say that $f$ is $\varepsilon$-calibrated with respect to a utility function $u$ if $\mathrm{UC}(f, u) \leq \varepsilon$. Note that for any $I = [a, b]$, the inner optimization problem in (3.2) can be rewritten as

$$\left|\mathbb{E}\left[u(f(X), Y) - u(f(X), \hat{Y})|v_u(X) \in [a, b]\right]\right| \mathbb{P}\{v_u(X) \in [a, b]\}.$$

In words, looking at the instances where $v_u(X) \in [a, b]$, the bias between the utility the decision-maker expects to get (while using $f(X)$ to take decisions and to estimate the utility) and the actual utility the decision-maker achieves (when using $f(X)$ to take decisions), is at most $\varepsilon$ after being weighted by the probability of the event $\{v_u(X) \in [a, b]\}$.

Combining (3.1) and (3.2) above, one obtains that $\mathrm{UC}(f, u)$ is equivalent to

$$\mathrm{UC}(f, u) = \sup_{I \in \mathbb{I}[-1,1]} |\mathbb{E}\left[\langle Y - f(X), \vec{u}(X)\mathbb{1}\{v_u(X) \in I\}\rangle\right]|. \tag{3.3}$$

Thus, utility calibration is equivalent to weighted calibration (2.3), with the witness class $\mathcal{W}$ set to $\mathcal{W}(u) := \{x \mapsto \xi\vec{u}(x)\mathbb{1}\{v_u(x) \in I\} | I \in \mathbb{I}[-1, 1]\}$. In addition, our notion of utility calibration requires that the predicted label $\hat{Y} \sim f(X)$ can be used for an unbiased estimation of the utility. This is related to Outcome Indistinguishability (OI) [32], where a predictor $f$ is considered reliable if its simulated outcomes $\hat{Y} \sim f(X)$ are computationally indistinguishable from Nature's true outcomes $Y$. We also note that this perspective also connects to recent work that leverages OI variants to establish links between loss minimization guarantees, omnipredictors, and multicalibration [33–35].

## 3.1 Decision-Theoretic Implications of Utility Calibration

In a very recent work, for the binary classification setting, Rossellini et al. [23] introduced the CutOff calibration metric, which assesses the calibration error by measuring against the worst-case bin, and demonstrated that it provide robust decision-theoretic guarantees. We defer a more detailed discussion of CutOff calibration to Appendix B.1. By assessing the $\mathrm{UC}(f, u)$ on the worst-case interval of $v_u(\cdot)$, our construction of utility calibration can be seen as a generalization of CutOff calibration to multiple dimensions and arbitrary utility functions, and that in fact inherits analogous decision-theoretic guarantees to the one shown in Rossellini et al. [23, Prop 2.1 and 3.2].

In particular, consider a decision rule based on thresholding the predicted utility $v_u(X)$ at some level $t_0 \in [-1, 1]$, i.e., taking the action $\hat{U}_{t_0} := \mathbb{1}\{v_u(X) \geq t_0\}$. This models the situation in which a user needs to commit a binary decision after estimating the utility using $f(X)$. Then, the quality of this decision can be assessed by the loss $\ell_{\mathrm{util}}(\tilde{u}, \hat{U}; t) = |\tilde{u} - t|\,\mathbb{1}\{\hat{U} \neq \mathbb{1}\{u \geq t\}\}$, which penalizes the *deviation* between the true utility $u_Y$ and the decision threshold $t_0$ when a mismatch between $\hat{U}_{t_0}$ and the ideal decision occurs. Consequently, let $R_{\mathrm{util}}(g; t_0) = \mathbb{E}[\ell_{\mathrm{util}}(u(f(X), Y), \hat{U}_{t_0}; t_0)]$ be the associated risk. Then, we show that the decision process $\hat{U}_{t_0}$ cannot significantly be improved by any simple post-processing of $v_u(\cdot)$ through a composition with a monotone function.

**Proposition 3.1** (Utility Risk Gap). *Let $u : \Delta^{C-1} \times \mathcal{Y} \to [-1, 1]$ be a utility function and $v_u(X)$ be the predicted expected utility. For any threshold $t_0 \in [-1, 1]$ and the loss function $\ell_{\mathrm{util}}$ as described above,*

$$R_{\mathrm{util}}(v_u(X); t_0) - \inf_{\substack{h:[-1,1]\to[-1,1] \\ \text{monotone}}} R_{\mathrm{util}}(h(v_u(X)); t_0) \leq 2\mathrm{UC}(f, u).$$

In words, Proposition 3.1 indicates that, if $f$ is utility calibrated, in such a binary decision-making scenario, the user can barely benefit from any monotonic post-processing to $v_u$. Another interpretation of $v_u(X)$ is as a regressor for the realized utility $u_Y := u(f(X), Y) \in [-1, 1]$. Similar to Rossellini et al. [23, Prop 2.1], we can show that the regressor $v_u$ satisfies a notion of calibration itself. First, note that distance from calibration naturally extends to such a single-dimension regression problem by considering a function $g_u(X)$ to be a perfectly calibrated predictor of $u_Y$ if $\mathbb{E}[u_Y \mid g_u(X)] = g_u(X)$ almost surely. We denote this extended notion of distance from calibration as $\mathrm{DCU}(f, u)$, the Distance to Calibrated Utility Predictor for $v_u(X)$ with respect to the realized utility $u(f(X), Y)$:

$$\mathrm{DCU}(f) := \inf_{\substack{g_u:\mathcal{X}\to[-1,1] \\ \mathbb{E}[u_Y|g_u(X)]=g_u(X)}} \mathbb{E}\,|g_u(X) - v_u(X)|\,.$$

We show that $\mathrm{DCU}(f, u)$ can be effectively controlled through $\mathrm{UC}(f, u)$.

**Proposition 3.2** (Utility Calibration upper Bounds DCU). *Let $u : \Delta^{C-1} \times \mathcal{Y} \to [-1, 1]$ be a utility function. Then,*

$$\mathrm{DCU}(f) \leq \sqrt{8\mathrm{UC}(f, u)} + \mathrm{UC}(f, u).$$

Proposition 3.2 implies that if $\mathrm{UC}(f, u)$ is small, then $v_u(X)$, seen as a regressor for the true utility $u(f(X), Y)$, is a calibrated predictor itself. This further strengthens the interpretation of $\mathrm{UC}(f, u)$: not only does it *ensure actionable decisions based on $v_u(X)$*, but it also *guarantees that $v_u(X)$ itself is not far from calibration*. We thus turn to the question of how to estimate $\mathrm{UC}(f, u)$.

## 3.2 Measuring $\mathrm{UC}(f, u)$

A naturally arising question is on the difficulty of measuring and achieving a small utility calibration error. We show in Lemma 3.3 that both the computational and sample complexity of estimating $\mathrm{UC}(f, u)$ are generally feasible and of limited dependence on the dimension, allowing its scalability to predictors with thousands of classes.

**Lemma 3.3** (Estimating Utility Calibration Against a Single Function). *Let $u : \Delta^{C-1} \times \mathcal{Y} \to [-1, 1]$ be a fixed utility function and $f : \mathcal{X} \to \Delta^{C-1}$ be a given predictor. Define the empirical estimator $\widehat{\mathrm{UC}}(f, u; S)$ based on $n$ i.i.d. samples $S = \{(X_i, Y_i)\}_{i=1}^n \sim D^n$ as*

$$\widehat{\mathrm{UC}}(f, u; S) := \sup_{I \in \mathbb{I}[-1,1]} \left| \frac{1}{n} \sum_{i=1}^n [(u(f(X_i), Y_i) - V(X_i))\mathbb{1}\{V(X_i) \in I\}] \right|.$$

Then, for any $\delta > 0$, with probability at least $1 - \delta$ over the draws of the sample $S$,

$$|\widehat{\mathrm{UC}}(f, u; S) - \mathrm{UC}(f, u)| \leq \tilde{O}\left(\sqrt{\frac{\log(1/\delta)}{n}}\right). \qquad (3.4)$$

Furthermore, $\widehat{\mathrm{UC}}(f, u; S)$ can be computed from $S$ in $O(n^2 + nT_{eval})$ time, where $T_{eval}$ is the time to evaluate $f(X_i)$ and $u(\cdot, \cdot)$.

First, we note that the constants hidden in the $\tilde{O}(\cdot)$ in (3.4) are dimension-independent. Similarly, the only dimension-dependent term in the computational complexity is $T_{eval}$. As such, $\mathrm{UC}(f, u)$ is a completely scalable notion of calibration, allowing it to be implemented for classifier with a thousand classes – as exemplified in Section 4. In addition, given that $\mathrm{UC}(f, u)$ can be formulated as weighted calibration (see eq. (3.3)) and that $\widehat{\mathrm{UC}}(f, u; S)$ is both a computationally and sample efficient, we can leverage the common patching-style post-hoc calibration algorithm, eg: [9, 36, 2] to recalibrate $f$ in order to minimize $\mathrm{UC}(f, u)$ while decreasing its Brier score. We summarize this fact informally in Lemma 3.4 and defer to a more detailed discussion and experimental evaluation of the recalibration patching algorithm in Appendix A.

**Lemma 3.4** (Informal). *For $\varepsilon > 0$, there exists an algorithm, which given a classifier $f : \mathcal{X} \to \mathcal{Y}$, outputs a recalibrated classifier $\tilde{f} : \mathcal{X} \to \mathcal{Y}$ such that $\mathrm{UC}(\tilde{f}, u) \leq \varepsilon$ and its Brier score decreases:*

$$\mathbb{E}\left[\|\tilde{f}(X) - Y\|_2^2\right] \leq \mathbb{E}\left[\|f(X) - Y\|_2^2\right].$$

Those encouraging facts on the utility calibration w.r.t. a single $u$ being established, we next turn out attention to Utility Calibration against a function classe $\mathcal{U}$.

## 3.3 Utility Calibration against a Function Class

In many real-world scenarios, a single probabilistic predictor $f$ might serve multiple downstream users, or a single user might employ it under varying conditions or objectives. The exact utility function relevant at the time of decision-making may not be known beforehand by the model provider, or it might even change over time (e.g., due to changing costs, available actions, or strategic goals), or might be fundamentally user-dependent.

Therefore, ensuring reliability often requires guarantees that hold not just for a single, pre-specified utility function, but for an entire class of plausible or relevant utility functions, denoted by $\mathcal{U}$. This provides a more robust assurance that the model's predictions are trustworthy across a range of potential downstream applications. To capture this requirement, overloading the notion, we define utility calibration against a function class as the worst-case performace over the class, i.e.

$$\mathrm{UC}(f, \mathcal{U}) = \sup_{u \in \mathcal{U}} \mathrm{UC}(f, u). \qquad (3.5)$$

To illustrate the practical relevance of this concept, we exhibit hereafter several examples of utility classes, each motivated by different downstream tasks. We first demonstrate how to recover similar notions to top-class (2.1) and class-wise (2.2) using the framework of utility calibration (3.5).

**Example 3.5** (Top-Class and Class-Wise Utilities ($\mathcal{U}_{\mathrm{TCE}}, \mathcal{U}_{\mathrm{CWE}}$)). *Define the top-class utility function $u_{\mathrm{top}}(p, y) = \mathbb{1}\{y = e_{\gamma(p)}\}$, where we recall that $\gamma(p) = \arg\max_k p_k$, and the class-wise utility function for class $c \in [C]$ as $u^c(p, y) = \mathbb{1}\{y = e_c\}$. The corresponding utility classes are respectively $\mathcal{U}_{\mathrm{TCE}} = \{u_{\mathrm{top}}\}$ and $\mathcal{U}_{\mathrm{CWE}} = \{u^c, \ c \in [C]\}$. It results in defining:*

$$\mathrm{UC}(f, \mathcal{U}_{\mathrm{TCE}}) = \sup_{I \in \mathbb{I}[0,1]} \left|\mathbb{E}\left[\left(\mathbb{1}\left\{Y = e_{\gamma(f(X))}\right\} - f(X)_{\gamma(f(X))}\right) \mathbb{1}\left\{f(X)_{\gamma(f(X))} \in I\right\}\right]\right|,$$

$$\mathrm{UC}(f, \mathcal{U}_{\mathrm{CWE}}) = \sup_{c \in [C]} \sup_{I \in \mathbb{I}[0,1]} \left|\mathbb{E}\left[\left(\mathbb{1}\left\{Y = e_c\right\} - f(X)_c\right) \mathbb{1}\left\{f(X)_c \in I\right\}\right]\right|.$$

In contrast to the binned estimators $\mathrm{TCE}^{\mathrm{bin}}$ (2.1) and $\mathrm{CWE}^{\mathrm{bin}}$ (2.2), utility calibration with the classes $\mathcal{U}_{\mathrm{TCE}}$ and $\mathcal{U}_{\mathrm{CWE}}$ offers a more robust, binning-free, computable assessment. Specifically, $\mathrm{UC}(f, \mathcal{U}_{\mathrm{TCE}})$ and $\mathrm{UC}(f, \mathcal{U}_{\mathrm{CWE}})$ are determined by maximizing the calibration deviation over *any* possible interval $I \subseteq [0, 1]$ (and additionally over classes for $\mathcal{U}_{\mathrm{CWE}}$), effectively identifying the

worst-case interval-based error. This approach inherently avoids fixed binning schemes, thereby circumventing pathologies where bin choices drastically alter estimated errors [8, 22]. Consequently, for any binning scheme using $m$ bins, $m \cdot \mathrm{UC}\,(f, \mathcal{U}_{\mathrm{TCE}})$ and $m \cdot \mathrm{UC}\,(f, \mathcal{U}_{\mathrm{CWE}})$ upper bound $\mathrm{TCE}^{\mathrm{bin}}(f)$ and $\mathrm{CWE}^{\mathrm{bin}}(f)$ respectively, while the converse is not true. We refer to Appendix B.2 for the formal statement. Furthermore, by Proposition 3.1, a small $\mathrm{UC}\,(f, \mathcal{U}_{\mathrm{TCE}})$ guarantees that decisions based on thresholding top-class confidence are robust to monotonic recalibration, and by Proposition 3.2 that this confidence is a calibrated predictor of actual top-class accuracy. Analogous guarantees hold for $\mathrm{UC}\,(f, \mathcal{U}_{\mathrm{CWE}})$ for individual class confidences, offering assurances for downstream applications.

Beyond the binarized perspectives offered by $\mathcal{U}_{\mathrm{TCE}}$ and $\mathcal{U}_{\mathrm{CWE}}$, the utility calibration framework readily accommodates richer and more complex classes of utility functions. This allows us to move beyond presumptive binary events and consider more nuanced downstream applications. In particular, consider settings where the utility derived from an outcome $Y$ is intrinsic to the outcome itself, independent of the model's prediction $f(X)$. For example, in medical diagnosis, the cost or severity tied to a specific disease $Y = e_j$ might be a fixed value $a_j$, irrespective of the diagnostic prediction. Formally, such situations can be modeled using a utility function $u_a : \Delta^{C-1} \times \mathcal{Y} \to [-1, 1]$ defined by a payoff vector $a \in [-1, 1]^C$, where utility function and the expected utility are respectively $u_a(\cdot, e_j) = a_j$ and $v_{u_a}(X) = \langle f(X), a \rangle$, with $a_j$ represents the utility if the true outcome is $e_j$.

**Example 3.6** (Linear Utilities ($\mathcal{U}_{\mathrm{lin}}$))**.** *Define the class of linear utilities as $\mathcal{U}_{\mathrm{lin}} := \{u_a \mid a \in [-1, 1]^C\}$, noting that the predicted utility $v_{u_a}(X)$ is linear in the prediction $f(X)$.*

A small $\mathrm{UC}(f, \mathcal{U}_{\mathrm{lin}})$ ensures that for any payoff vector $a$, the predicted expected utility $v_{u_a}(X)$, as a regressor of the realized utility, is close to calibration.

Alternatively, in applications like information retrieval or recommender systems, the realized utility depends on the rank assigned to the true outcome $Y = e_j$. Given a model's prediction $p = f(X)$, assuming $p_1, \ldots, p_C$ are distinct (or that ties are broken arbitrarily/randomly among equal coordinates), the rank of class $j$, denoted $\mathrm{rank}(p, j)$, is its position across $p$, i.e. $\mathrm{rank}(p, j) := \sum_{i \in [C]} \mathbb{1}\{p_j \leq p_i\}$. Using a valuation vector $\theta \in [-1, 1]^C$, a rank-based utility function can then be constructed as $u_\theta(p, e_j) = \theta_{\mathrm{rank}(p,j)}$ with the associated expected utility function $v_{u_\theta}(X) = \sum_{i=1}^C f(X)_i \theta_{\mathrm{rank}(f(X),i)}$. Calibrating for such utilities ensures the model's expected rank-based performance aligns with reality. A prominent special case is $\mathrm{topK}$ utility, where the valuation vector $\theta^{(K)}$ for a given $K \in [C]$ is defined such that $\theta_r^{(K)} = 1$ if $r \leq K$ and $\theta_r^{(K)} = 0$ if $r > K$.

**Example 3.7** (Rank-Based and Top-$K$ Utilities ($\mathcal{U}_{\mathrm{rank}}, \mathcal{U}_{\mathrm{topK}}$))**.** *The class of general rank-based utilities is $\mathcal{U}_{\mathrm{rank}} := \{u_\theta \mid \theta \in [-1, 1]^C\}$. The class of top-$K$ utilities is then $\mathcal{U}_{\mathrm{topK}} := \{u_{\theta^{(K)}} \mid K \in [C]\}$, where $\theta_r^{(K)} = \mathbb{1}(r \leq K)$. Equivalently, $u_K(p, e_j) = \mathbb{1}\{\mathrm{rank}(p, j) \leq K\}$. A small $\mathrm{UC}(f, \mathcal{U}_{\mathrm{rank}})$ (or $\mathrm{UC}(f, \mathcal{U}_{\mathrm{topK}})$) ensures reliable prediction for general rank (or specifically top-$K$ accuracy) valuations, validating the model's ranking capabilities.*

As discussed in Section 2, decision calibration [14] ensures that for problems with up to $K$ actions, the model's predicted utility for its recommended action matches the actual realized utility. We can frame a similar guarantee within utility calibration. For any bounded loss function $l : \mathcal{Y} \times [K] \to [-1, 1]$ and a prediction $p = f(X)$, the optimal action is $\delta_l(p) = \arg\min_{a \in [K]} \mathbb{E}_{\hat{Y} \sim p}[l(\hat{Y}, a)]$. The utility function is then $u_l(p, y) = -l(y, \delta_l(p))$, representing the negative loss from outcome $y$ under action $\delta_l(p)$. The predicted expected utility is $v_{u_l}(X) = -\mathbb{E}_{\hat{Y} \sim f(X)}[l(\hat{Y}, \delta_l(f(X))) | X]$.

**Example 3.8** (Decision Calibration Utilities ($\mathcal{U}_{\mathrm{dec},K}$))**.** *Let $\mathcal{L}_K = \{l : \mathcal{Y} \times [K] \to [-1, 1]\}$ be the class of all bounded $K$-action loss functions, and the utility class is $\mathcal{U}_{\mathrm{dec},K} := \{u_l, l \in \mathcal{L}_K\}$. A small $\mathrm{UC}(f, \mathcal{U}_{\mathrm{dec},K})$ implies that for any $K$-action decision problem $l \in \mathcal{L}_K$, the model's prediction of expected utility for its chosen action $\delta_l(f(X))$ reliably reflects the achieved utility $-l(Y, \delta_l(f(X)))$.*

These aforementioned examples illustrate that calibrating against classes $\mathcal{U}$ provides guarantees tailored to diverse user needs, moving beyond simplistic binarized assessments. A critical question then arises: how can $\mathrm{UC}\,(f, \mathcal{U})$ be measured for a given class $\mathcal{U}$, which we address in the next section.

### 3.4 Measurability of utility calibration

Estimating $\sup_{u \in \mathcal{U}} \mathrm{UC}\,(f, u)$ in (3.5) presents two key challenges: the *computational complexity* of the optimization, and the *sample complexity* required for the empirical supremum to converge to its

true value. We introduce the two notions of proactive and interactive measurability to decouple these two aspects.

**Definition 3.9** (Proactive Measurability). *The utility calibration error w.r.t. class $\mathcal{U}$ is proactively measurable if there exists an algorithm $A$ and polynomial functions $N_{poly}, T_{poly}$ duch that for any $\varepsilon, \delta > 0$ and $n \geq N_{poly}(C, 1/\varepsilon, 1/\delta)$ samples $S \sim D^n$, algorithm $A(S)$ outputs $\hat{u}$ satisfying $|\mathrm{UC}(f, \hat{u}) - \mathrm{UC}(f, \mathcal{U})| \leq \varepsilon$ with probability at least $1 - \delta$ and the runtime of $A(S)$ is bounded by $T_{poly}(C, n)$.*

Generally, for a finite class $\mathcal{U}$, if $|\mathcal{U}|$ grows polynomially in $C$ then by Lemma 3.3 we can guarantee proactive measurability. Nonetheless, even for simple infinite classes such as $\mathcal{U}_{\mathrm{lin}}$, proactive measurability reduces to a non-convex optimization problem that cannot be generally solved in polynomial time. In fact, even aiming for a weaker notion, namely *improper auditing*, Gopalan et al. [11] showed that assessing both weaker and stronger notions than $\mathrm{UC}(f, \mathcal{U}_{\mathrm{lin}})$ cannot be done in polynomial time in both the error $\varepsilon^{-1}$ and the dimension $C$ [11, Theorem 1.3, Theorem 5.2, and Theorem 8.6]. A more detailed description of Gopalan et al. [11] hardness results is in Appendix B.3. The primary bottleneck is the *computation time*. Next, we thus propose an alternative criteria of measurability that decouples the statistical guarantee from the computational complexity of verifying the supremum.

**Definition 3.10** (Interactive Measurability). *The utility calibration error w.r.t. class $\mathcal{U}$ is interactively measurable if there exists an estimator $\widehat{\mathrm{UC}}(f, u; S)$ and a polynomial function $N_{poly}$ such that for $n \geq N_{poly}(C, 1/\varepsilon, 1/\delta)$ samples $S \sim D^n$, it holds with probability at least $1 - \delta$ that*

$$\sup_{u \in \mathcal{U}} |\widehat{\mathrm{UC}}(f, u; S) - \mathrm{UC}(f, u)| \leq \varepsilon.$$

Interactive measurability represents a much more achievable goal. For example, while decision calibration is computational hard to measure, Zhao et al. [14] showed that it admits polynomial sample complexity. In Appendix B.4, we further demonstrate the interactive measurability of different utility classes of interest with controlled Rademacher complexity.

In summary, while proactively measuring the worst-case utility calibration error $\mathrm{UC}(f, \mathcal{U}) = \sup_{u \in \mathcal{U}} \mathrm{UC}(f, u)$ is often computationally prohibitive for expressive utility classes $\mathcal{U}$, interactive measurability allows for efficient estimation of $\mathrm{UC}(f, u)$ uniformly for any *specific* $u \in \mathcal{U}$. Next, we leverage this distinction to propose a scalable evaluation methodology that, instead of pursuing the intractable worst-case error, characterizes the *distribution* of utility calibration errors across $\mathcal{U}$. This provides a more nuanced understanding of a model $f$'s calibration reliability over a spectrum of potential downstream applications, that we then evaluate in experiments.

# 4 Scalable Evaluation of Utility Calibration and Experiment

**Scalable Evaluation of Utility Calibration.** Our approach considers a probability distribution $\mathcal{D}_{\mathcal{U}}$ over the utility class $\mathcal{U}$. Many utility classes of interest admit a finite-dimensional parameterization, making sampling from $\mathcal{D}_{\mathcal{U}}$ practical. We sample $M$ utility functions $\{u_m\}_{m=1}^M$ from $\mathcal{D}_{\mathcal{U}}$ and, for each $u_m$, compute its estimated error $\widehat{E}_{m,n} := \widehat{\mathrm{UC}}(f, u_m; S)$ using $n$ data points from a sample $S$. These $M$ error estimates then form an *empirical Cumulative Distribution Function (eCDF)*,

$$\widehat{F}_{E,M,n}(e) := \frac{1}{M} \sum_{m=1}^M \mathbb{1}\{\widehat{E}_{m,n} \leq e\},$$

which serves as an empirical proxy for the true CDF, $F_E(e) := \mathbb{P}_{u \sim \mathcal{D}_{\mathcal{U}}}(\mathrm{UC}(f, u) \leq e)$. We provide guarantees on the difference between $F_E(e)$ and $\hat{F}_{E,M,n}(e)$ in Appendix B.5.

In particular, $\mathcal{U}_{\mathrm{lin}}$ (Example 3.6) and $\mathcal{U}_{\mathrm{rank}}$ (Example 3.7) both admit finite-dimension parameterization. For $\mathcal{U}_{\mathrm{lin}}$, we construct $\mathcal{D}_{\mathcal{U}_{\mathrm{lin}}}$ by sampling the payoff vectors $a$ uniformly in $[-1, 1]^C$. Meanwhile, for $\mathcal{U}_{\mathrm{rank}}$, we also sample from $\mathcal{D}_{\mathcal{U}_{\mathrm{rank}}}$ by uniformly sampling valuation vectors $\theta \in [-1, 1]^C$, which satisfy $\theta_1 \geq \theta_2 \geq \cdots \geq \theta_C$. This is to reflect a rational preference for better ranks, i.e. the higher the rank of the true realization within the predictions of $f(X)$, the higher the utility.

**Numerical experiments.** We now demonstrate how our approach can be used to empirically validate model calibration. For all of our experiments, we used pretrained models for ImageNet and CIFAR10/100 [37, 38]. In Appendix D, we further detail our experimental setup, provide additional results, and list the licenses of all the assets used. Here, we present the results of two

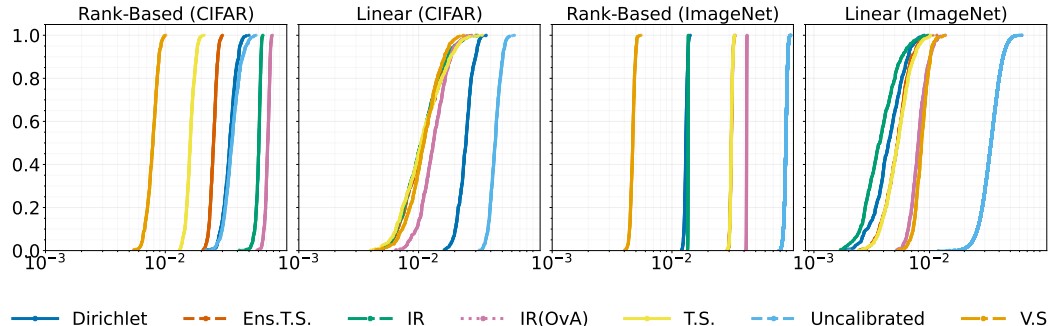

Figure 1: eCDF of utility calibration errors for `ResNet20` on CIFAR100 (left two panels) and `ViT` on ImageNet-1K (right two panels).

settings: (1) `ResNet20` [39] on CIFAR100 and a Vision Transformer `ViT` [40] on ImageNet-1K. For post-hoc calibration, we applied Temperature Scaling (T.S.) [26], Vector Scaling (V.S.) [41], Ensemble Temperature Scaling (Ens. T.S.) [42], and Dirichlet recalibration [27]. In addition, we fitted a shared Isotonic Regression (I.R.) [28] across different classes and an Isotonic Regression for each class using one-vs-all approach (IR OvA).

In Table 1, we present a detailed comparison for the `ResNet20` model on CIFAR100. This table compares standard metrics (accuracy, Brier score), binned binarized metrics ($\text{TCE}_{\text{binned}}$, $\text{CWE}_{\text{binned}}$ with 15 equal-weight bins), and our utility calibration metrics for specific utility classes: top-class ($\mathcal{U}_{\text{TCE}}$), class-wise ($\mathcal{U}_{\text{CWE}}$), and top-$K$ ($\mathcal{U}_{\text{TopK}}$). As expected, most post-hoc methods improve Brier scores and reduce binned error over the uncalibrated model, often with minimal accuracy impact. Our binning-free utility calibration metrics, $\mathcal{U}_{\text{TCE}}$, $\mathcal{U}_{\text{CWE}}$, and $\mathcal{U}_{\text{TopK}}$, show similar improvements. Notably, while $\mathcal{U}_{\text{TCE}}$ and $\mathcal{U}_{\text{TopK}}$ are equal for the uncalibrated model, they can diverge for calibrated models. Since $\mathcal{U}_{\text{TopK}}$ considers all $K \in [C]$, it upper-bounds $\mathcal{U}_{\text{TCE}}$ (the $K = 1$ case). Although calibration methods reduce $\mathcal{U}_{\text{TCE}}$ effectively, the typically higher $\mathcal{U}_{\text{TopK}}$ values can reveal miscalibration for ranks beyond top-1. This suggests $\mathcal{U}_{\text{TopK}}$ as a more comprehensive benchmark.

Beyond specific utility functions, Figure 1 displays the eCDFs of utility calibration errors for broader utility classes: rank-based ($\mathcal{U}_{\text{rank}}$) and linear ($\mathcal{U}_{\text{lin}}$). Each eCDF, generated from $M = 1000$ sampled utility functions, shows the proportion of utilities for which the calibration error is below a certain threshold; thus, curves shifted to the left indicate superior calibration across the wider class of utility functions. For the `ResNet20` on CIFAR100 (left panels), the eCDFs reveal interesting dynamics. While most post-hoc methods improved upon the uncalibrated model for $\mathcal{U}_{\text{lin}}$, some methods, specifically I.R. and I.R.(OvA), surprisingly worsened performance for $\mathcal{U}_{\text{rank}}$ compared to the uncalibrated model. This degradation was not apparent from the specific metrics in Table 1, underscoring the necessity of the broader perspective offered by these eCDF plots across a class of utilities. For the Vision Transformer (`ViT`) on ImageNet-1K (right panels), the uncalibrated model exhibits the poorest performance across both $\mathcal{U}_{\text{rank}}$ and $\mathcal{U}_{\text{lin}}$. Nevertheless, the eCDF plots still provide a nuanced way to compare and evaluate different post-hoc methods against each other.

In conclusion, utility calibration provides a robust, unified, and application-centric framework for evaluating classifier reliability. Its specific instantiations, $\mathcal{U}_{\text{CWE}}$ and $\mathcal{U}_{\text{TCE}}$, offer superior, binning-free alternatives to traditional metrics with actionable guarantees, while $\mathcal{U}_{\text{TopK}}$ presents an even more comprehensive ranking assessment. Furthermore, the eCDF plots across broader utility classes deliver crucial nuanced insights into model behavior that single-metric evaluations obscure.

Table 1: `ResNet20`-CIFAR100 calibration results. Comparison of post-hoc methods using Accuracy, binned ECEs ($\text{TCE}_{\text{eqBin}}$, $\text{CWE}_{\text{eqBin}}$), and utility calibration errors: $\mathcal{U}_{\text{TCE}}$ (Top-Class), $\mathcal{U}_{\text{CWE}}$ (Class-Wise), $\mathcal{U}_{\text{topK}}$ (Top-K). Mean $\pm$ maximum deviation over 5 splits.

| Method | Accuracy | Brier Score | $\text{CWE}_{\text{binned}}$ | $\text{TCE}_{\text{binned}}$ | $\mathcal{UC}_{\text{CWE}}$ | $\mathcal{UC}_{\text{TCE}}$ | $\mathcal{UC}_{\text{TopK}}$ |
|---|---|---|---|---|---|---|---|
| Uncalibrated | $0.677 \pm 0.010$ | $0.480 \pm 0.015$ | $0.00214 \pm 0.00016$ | $0.1600 \pm 0.008$ | $0.0124 \pm 0.0011$ | $0.1590 \pm 0.015$ | $0.1590 \pm 0.015$ |
| Dirichlet | $0.666 \pm 0.010$ | $0.457 \pm 0.008$ | $0.00194 \pm 0.00014$ | $0.0727 \pm 0.0160$ | $0.0111 \pm 0.0004$ | $0.0709 \pm 0.0165$ | $0.0818 \pm 0.0154$ |
| IR | $0.677 \pm 0.010$ | $0.444 \pm 0.011$ | $0.00186 \pm 0.00006$ | $0.0264 \pm 0.0033$ | $0.0113 \pm 0.0005$ | $0.0310 \pm 0.0071$ | $0.0756 \pm 0.0086$ |
| IR (OvA) | $0.674 \pm 0.010$ | $0.454 \pm 0.011$ | $0.00156 \pm 0.00016$ | $0.0454 \pm 0.0103$ | $0.0108 \pm 0.0011$ | $0.0467 \pm 0.0190$ | $0.0927 \pm 0.0091$ |
| T.S. | $0.677 \pm 0.010$ | $0.440 \pm 0.014$ | $0.00188 \pm 0.00008$ | $0.0250 \pm 0.0066$ | $0.0114 \pm 0.0005$ | $0.0322 \pm 0.0090$ | $0.0367 \pm 0.0046$ |
| Ens.T.S | $0.677 \pm 0.010$ | $0.440 \pm 0.010$ | $0.00196 \pm 0.00006$ | $0.0212 \pm 0.0045$ | $0.0114 \pm 0.0005$ | $0.0304 \pm 0.0063$ | $0.0393 \pm 0.0056$ |
| V.S. | $0.680 \pm 0.010$ | $0.435 \pm 0.010$ | $0.00150 \pm 0.00010$ | $0.0334 \pm 0.0117$ | $0.0107 \pm 0.0010$ | $0.0375 \pm 0.0148$ | $0.0403 \pm 0.0121$ |

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
