# OpenReview forum: "Scalable Utility-Aware Multiclass Calibration"
_NeurIPS.cc/2025/Conference — Submitted to NeurIPS 2025_

### Official Review · Reviewer_GWKc · 2025-06-19

**Clarity:** 3
**Significance:** 2
**Originality:** 3
**Rating:** 5
**Confidence:** 2

**Summary:**

Calibration evaluation is a difficult topic and existing metrics suffer from several issues (too specific like top-class, or computationally challenging). This work proposes a general framework to evaluate calibration according to a user-defined utility function. After being theoretically derived, the new metrics are applied to assessing the calibration of pre-trained models on image datasets, including after following post-hoc calibration.

**Questions:**

1. What is the impact of the test set size on the evaluation of the proposed metrics? For example by plotting averages and stds for a growing number of testing samples per class.

**Ethical Concerns:**

["NO or VERY MINOR ethics concerns only"]

**Final Justification:**

I find that the study of different calibration notions (top-class, class-wise, decision…) is usually disparate so I appreciate work towards unification, which this paper does, even though the experiments are quite limited.

**Limitations:**

limitations addressed in the appendix

**Paper Formatting Concerns:**

no major concern

**Quality:**

3

**Strengths And Weaknesses:**

**Strengths**

1. Well written, rigorous, and theoretically sound paper (math not carefully checked).
2. Instead of proposing a new notion of calibration hard to compare with existing ones, "utility calibration" unifies existing notions.
3. Scalability (experimentally demonstrated with up to 1000 classes).

**Weaknesses**

1. Despite studying calibration in an application-centric view ("through the lens of downstream applications" (line 7), "provide an application-focused perspective on calibration" (line 40), "many real-world scenarios"(line 233), and "practical relevance" (line 243)), most of the paper is only theoretical and do not provide practical advice on how to use utility calibration on practical settings. Unfortunately, this paper fails to bridge the gap between theory and practice in my opinion.
2. Experiments are limited: only image datasets and a couple of models. In my opinion, the experiments would benefit from testing more datasets and models.

---

> ### Author Rebuttal · Authors · 2025-07-30
>
> We thank the reviewer for their detailed feedback and positive assessment. We address the reviewer's points .
>
> **1. On the gap between theory and practice:**
>
> We thank the reviewer for this important comment. We agree that our paper does not provide prescriptive advice on how to design the optimal utility function for a specific application, such as in a particular biomedical context. Our goal is different but, we believe, equally practical.
>
> Our framework's practicality lies in providing a robust and scalable toolkit for the crucial meta-task of *benchmarking and comparing models*. Practitioners are often faced with choosing between different models or post-hoc calibration methods, and existing metrics can be misleading (for example, due to binning artifacts) or computationally infeasible for high-dimensional problems. Our contribution is to provide a practical solution to *this* problem, while enabling the practitioner to **choose the utility class**, e.g. among the examples given in Sec 3.3. Our ``application-centric'' claim stems from the ability that we offer to choose the class of utilities - a choice that is certainly application dependent, and can only be performed by a domain expert.
>
> For example, using our eCDF evaluation methodology (Section 4), across a whole class of potential downstream utilities (e.g., Rank-based in Fig. 1a and Linear in Fig. 1b),  a user can rigorously compare many models (TS, Dirichlet, IR, VS, etc.), gaining a much deeper understanding of their relative reliability, **for that specific choice of utilities** (a choice that once again will depend on the application). In this sense, we believe our work is practical, not at the level of domain-specific consultation, but at the crucial level of model development, evaluation, and selection. We will clarify this positioning in the updated version of the paper.
>
> > However, we noticed that reviewer GhnP was puzzled by the use of  "application centric" - we can rephrase as  "application-dependent'', or more systematically `"utility-aware'' (as in the title) if the reviewer feels it makes the contribution clearer.
>
> **2. On the scope of experiments:**
>
> We appreciate the suggestion to explore more diverse domains. Our initial focus on image classification, particularly with CIFAR-100 and ImageNet-1K, was driven by the need to test our framework's scalability on problems with a large number of classes (C=100 and C=1000, respectively), for which numerous high-quality pre-trained models are readily available.
>
> To demonstrate the versatility of our framework beyond computer vision, we commit to adding a new experiment on a text classification task in the final version of the paper. We plan to fine-tune a pre-trained BERT-based model on a standard text benchmark and apply our full utility calibration analysis.
>
> **3. On the impact of test set size:**
>
> This is an excellent question. Our theoretical analysis in **Lemma 3.3** shows that our empirical estimator for a given utility function converges to the true value at a rate of $\tilde{O}(1/\sqrt{n})$, where $n$ is the test set size. The experiments in our paper use several thousand test samples, which, according to our theory, is a regime that ensures accurate estimation.
>
> Nevertheless, the reviewer raises a very interesting point about the estimator's empirical behavior in lower-data regimes. We agree that investigating the estimator's bias and variance as a function of sample size is a valuable analysis for practitioners. To address this, we will add a new study to the appendix in the final version of the paper. In this study, we will vary the test set size and measure the stability (mean and standard deviation) of our metrics across multiple random subsamples.
>
> Again, we thank you for your feedback and we are confident that addressing them will improve the manuscript. We remain available to answer additional questions.

---

> > ### Comment · Reviewer_GWKc · 2025-08-02
> > **Response to rebuttal**
> >
> > Thank you for this clear rebuttal.
> >
> > 1. I now understand better the positioning.
> >
> > 2. Indeed, for image classification many pre-trained models exits, and yet the evaluation only contains a couple. Calibration behaviour differs for different models and the current evaluation setting is not sufficient in my opinion. Adding text classification would also be interesting.
> >
> > 3. Thank you for adding this study in the final version of the paper.
> >
> > Overall, I appreciate the effort to unify different calibration notions in one framework where they become “utility functions” and maintain my score.

---

### Official Review · Reviewer_RjtA · 2025-07-03

**Clarity:** 3
**Significance:** 3
**Originality:** 4
**Rating:** 4
**Confidence:** 2

**Summary:**

This paper presents a thoughtful and well-structured general framework for evaluating model calibration for multiclass setting by introducing a utility-aware approach that generalizes existing calibration measures. It is grounded in solid theoretical analysis, clearly discussing the limitations of standard binning-based metrics and proposing a principled alternative based on the classes of utility functions. The unified perspective on calibration through the lens of downstream utility functions is both theoretically sound and practically important. The authors further present a concise experimental evaluation and analysis based on the proposed framework

**Questions:**

1. Could you clarify the practical implementation details of the top‑k utility functions, including their parameters $r$ and the sampling procedure used?
2. Could you provide a failure case study where standard calibration measures fall short compared to utility-aware measures?

**Ethical Concerns:**

["NO or VERY MINOR ethics concerns only"]

**Final Justification:**

After considering the authors rebuttal and clarifications, I am satisfied that my main concerns have been adequately addressed.

While the evaluation section could still benefit from more extensive real-world case studies, the presented results and discussion convincingly illustrate the utility and necessity of the proposed approach. Overall, the work is technically solid, addresses an important gap in calibration evaluation for multi-class settings.

**Limitations:**

No, but it is discussed in the Paper Checklist

**Quality:**

4

**Strengths And Weaknesses:**

**Strengths:**
1. The paper provides a comprehensive introduction and background that clearly motivate the need for calibration metrics that assess calibration quality through the lens of downstream applications.
2. The authors proposed a general framework and provide thorough theoretical justifications, including guarantees and bounds for the proposed utility calibration error, as well as its relationship to traditional metrics.
3. The proposed empirical methodology using the eCDF over sampled utilities offers a richer analysis of calibration behavior across different contexts
4. The paper present examples and brief discussion for each standard utility function, which helps ground the abstract framework in practical applications.

**Weaknesses:**
1. Some concepts in the utility sampling procedures for top‑k utility are not fully specified. Specifically, what is the parameter $r$, and how are utilities sampled in this case?
2. While the authors offer theoretical runtime guarantees, it would be also helpful to present empirical runtime to validate these claims experimentally.
3. The experimental section, although demonstrating the application of the proposed metrics, lacks depth in analysis. For example, the discussion of Table 1 does not provide clear insights. While eCDF plots reveal interesting calibration trends, the reasons behind these trends are not discussed. Additionally, discussion of failure cases and practical recommendations is lacking. It would be helpful to understand when standard metrics failure and when the proposed utility-aware approach is essential.

---

> ### Author Rebuttal · Authors · 2025-07-30
>
> We thank the reviewer for their thoughtful feedback and positive assessment of our work. We appreciate the constructive questions, which we proceed to answer below.
>
> **Response to Q1 and Weakness 1: Clarification on Top-K Utility**
>
> As defined in Example 3.7, the class of top-K utilities, ${U_{TopK}}$, is a *finite set of $C$ distinct utility functions*:  ${U_{TopK}} = \lbrace u_K \mid K \in [C]\rbrace$, where $u_K(p, e_j)$ is 1 if the rank of class $j$ for prediction $p$ is less than or equal to $K$, and 0 otherwise. This can be seen as a generalization of top-class calibration.
>
> The metric reported in our tables as $UC_{TopK}$ is the *worst-case utility calibration error over this entire finite set*: $UC(f,  U_{TopK}) = \sup_{K \in [C]} UC(f, u_K)$. Since this is a supremum over a finite set of size $C$, we can compute it directly by evaluating $UC(f, u_K)$ for each $K=1, \dots, C$ and taking the maximum. Thus, no sampling is required for this metric.
>
> In contrast, the sampling-based eCDF evaluation described in Section 4 is used for the much richer, infinite utility classes like linear utilities, U_lin, and general rank-based utilities, U_rank. For these classes, computing the true supremum is computationally intractable, making the eCDF approach a practical and powerful tool for analysis. We will revise the text to make this distinction clearer.
>
> **Response to Weakness 2: Empirical Runtime**
>
> We thank the reviewer for the suggestion to discuss empirical runtime. While we did not include a dedicated table, we can confirm that our evaluation framework is highly scalable. The total compute for all experiments was approximately 80 GPU hours, but the vast majority of this was spent on fitting the various baseline post-hoc calibration methods.
>
> Our evaluation framework itself is very efficient. For a given model, generating a full eCDF plot (by sampling 1000 utilities and evaluating the error for each) took approximately 60 minutes. This performance was achieved using a parallelized JAX implementation, which we will open-source.
>
> **Response to Weakness 3 and Q2: Deeper Experimental Analysis and Failure Cases**
>
> We appreciate the reviewer's push for a deeper analysis and are happy to elaborate on the insights from our framework.
>
> **Deeper Insights from Our Metrics:** Our utility-aware metrics offer a more discerning analysis than standard ones. For instance, in Table 1 (ResNet20-CIFAR100), while Isotonic Regression (IR) and Temperature Scaling (T.S.) show similar improvements on top-class error (our UC_TCE metric), our more comprehensive UC_TopK metric reveals a key difference: T.S. calibrates well across all top-K ranks, whereas IR's benefit is largely confined to the top-1 prediction. In this case, UC_TopK provides a more complete picture of calibration reliability.
>
> **Failure Case for Standard Metrics and Practical Implications:** Our eCDF analysis provides an example of a failure case where standard metrics can be misleading. As seen in Figure 1, some post-hoc methods perform *worse* than the uncalibrated model for the class of rank-based utilities, U_rank, even as standard metrics like binned TCE suggest improvement. This critical trade-off is not observable when looking at classical metrics alone.
>
> This analysis underscores the practical takeaway from our work. By providing tools to evaluate calibration against specific utility classes or to visualize performance distributions via eCDFs, our framework empowers practitioners to make more informed choices about model reliability.
>
> Again, we thank the reviewer for their valuable and constructive feedback. We are eager to answer any additional questions during the discussion period.

---

> > ### Comment · Reviewer_RjtA · 2025-08-04
> >
> > I thank the authors for the thoughtful rebuttal. Their response adequately addresses my concerns, and I have no remaining questions. In light of these, I maintain my positive assessment.

---

### Official Review · Reviewer_iTGh · 2025-07-03

**Clarity:** 2
**Significance:** 2
**Originality:** 2
**Rating:** 4
**Confidence:** 4

**Summary:**

The paper proposes a framework for efficiently measuring/assessing calibration of models in a multiclass setting. The proposed framework is downstream-utility-centric in that its definition of calibration measures how faithfully downstream utilities are estimated when the model's predictions are used instead of the true labels, with respect to each utility in a class of downstream utilities.

The paper showcases how several variants of calibration that have appeared before --- such as top-class calibration on the classical side but also recent variants such as CutOff calibration --- are instances thereof. It starts off by proving some decision-theoretic benefits of optimizing for utility calibration.

Further, it mainly focuses on the perspective of tractability of estimation/assessment of calibration error, and towards that end shows that for a single utility the error metric is estimable, and that it also is, in an "interactive" way, for an entire class U of utilities.

Last but not least, it provides simulations of the proposed evaluation of utility calibration and thus demonstrates its benefits (such as the binning-free nature) vs. baselines with respect to two canonical utility families.

**Questions:**

To briefly distill from the above, I'd like to request the following:

(1) Please provide more experiments examining sensitivity to various utility classes beyond linear and ranked ones. In particular, natural questions --- like what happens in terms of convergence speed of the methods when the family of utilities consists of misaligned utilities vs. when it consists of aligned utilities --- will be good to visualize. More broadly, per-utility evaluations for small-ish nonlinear classes U appear of interest.

(2) Please provide a comparative discussion with mentioned literature.

(3) Please further improve presentation to better disentangle, in the reader's eye, the main focus --- i.e. the calibration assessment discussion --- from (post-hoc) calibration enforcement.

**Ethical Concerns:**

["NO or VERY MINOR ethics concerns only"]

**Final Justification:**

The authors gave a detailed response to my rebuttal (and to the other reviewers, whose discussions I have read carefully), addressing several points of concern. The point regarding extra experiments was covered in a synthetic aligned/misaligned utilities experiment (and I would welcome more such synthetic experiments with different utility functions for the camera ready). Furthermore, for the presentation merits of the paper my concern was that the evaluation aspect of the contribution was not properly emphasized throughout. In this sense, it is important to highlight the authors' willingness to refocus the presentation around the evaluation aspects of calibration (including the title).

On the flip side of my assessment of the contribution, I didn't necessarily find the theory that innovative compared to prior works, as articulated in the original review. Moreover, I agree with the opinion of the other reviewers that more "real-world" experiments would have been nice since the paper focuses on assessment. So in that way, I did not find the paper to be very innovative or stellarly well empirically executed, but it is a good-quality contribution nonetheless. I thus maintained my overall assessment of the paper after the rebuttal.

**Limitations:**

Yes.

**Quality:**

2

**Strengths And Weaknesses:**

The paper does a nice job systematizing, and expanding on, the recent discussion in the calibration and multicalibration literatures around tractability of high dimensional variants of calibration. It revisits some of the older, multiclass-to-binary reductions as well as some newer formulations, and shows how to instantiate them into a simple utility calibration framework.

Furthermore, they run a respectable number of experiments to confirm that controlling the proposed utility calibration error is indeed possible in some fundamental cases, and leads to nice binning-free approaches.

On the other hand, beyond the formulation of the framework and its properties and the empirical evaluations, I found the paper to be somewhat "everyday" on the technical side, with proof techniques mostly borrowing fairly directly from the multicalibration literature or from concrete related sources (such as Rossellini et al.'s recent paper in related subsection). While some concepts such as proactive and interactive measurability are interesting as presentation devices, they do not necessarily result in an entirely new perspective on estimation/auditing (e.g. proactive measurability is, roughly speaking, going to be too hard whenever the setting is hard, without many interesting regimes in-between; the situation is slightly more interesting with interactive measurability).

Moreover, I believe the novelty of the "utility calibration" concept itself to be fairly limited. Decision calibration being the first reference work on this topic that I am aware of, since that paper there have been further ones not cited or discussed here. The paper (https://arxiv.org/abs/2310.17651) defined a high-dimensional relaxation of calibration that is used towards obtaining (in the online adversarial setting) downstream no-regret guarantees for many decision makers with their utilities and requires just per-coordinate unbiasedness on events that are determined by the utility functions; this has since been extended to handle nonlinear classes of utilities (and in certain cases even all utilities such as https://arxiv.org/abs/2402.08753). The former paper's (https://arxiv.org/abs/2310.17651)  framework similarly to the current manuscript notes that it subsumes, and extends, top-class calibration; and their objective per each utility appears quite similar --- so a thorough discussion of the innovations in the framework proposed here appears necessary.  Other followup work to that paper has included a paper that explicitly studies the properties of Calibration errors for decision making (https://arxiv.org/abs/2404.13503); and further work that is related to (feature-less) omniprediction and also puts downstream utility at the center is U-calibration (https://arxiv.org/abs/2307.00168).

In addition, I was a bit confused by the paper declaring in a variety of places that its main focus is assessment of calibration metrics, but being relatively unfocused in that regard and interspersing the discussion with a variety of results showcasing e.g. utility benefits of the calibration method, its special instances, the patching-based post-hoc calibration method, u.a. The title also doesn't make it sufficiently clear that scalability is intended to refer primarily to evaluation. In that sense, the presentation of the paper appears somewhat unfocused/worth improving.

---

> ### Author Rebuttal · Authors · 2025-07-30
>
> We are grateful for the reviewer's feedback. We believe that integrating it will significantly strengthen our manuscript. We address the reviewer's questions and suggestions below.
>
> **Q1: Experiments on Sensitivity to Utility Class Structure**
>
> We thank the reviewer for this interesting suggestion. Following up on your question, and to better understand the dynamics of calibrating against different utility structures, we are currently conducting new experiments focused on the convergence speed of our patching algorithm (along with the performance) when faced with "aligned" versus "misaligned" non-linear utility classes. This experiment will provide another example of how the choice of the utility class $\mathcal{U}$ affects calibration. We hereafter detail this setup.
>
> More precisely, our additional utility model is based on a user-defined **gain matrix**  $R \in [0,1]^{C \times C}$, where $R_{ij}$ is the reward for predicting class $j$ when the true class is $i$. We set $R_{ii}=1$. Given a model's prediction $p$, a rational agent first chooses the action $j^\star(p)$ that maximizes their expected gain: $j^\star (p) = argmax_j \sum_i p_i R_{ij}$. The realized utility is then $u_R(p, y=e_i) = R_{i, j^\star(p)}$.
>
> We constructed two distinct utility classes based on this model:
> *   **Aligned Class:** This models a group of users with a similar, low tolerance for any error. A gain matrix $R$ for this class has $R_{ii}=1$ and all off-diagonal entries ($i \neq j$) sampled independently from a $\text{Uniform}(0, 0.1)$ distribution.
> *   **Misaligned Class:** This models a mix of specialists with distinct biases. We partition the classes into disjoint subsets ($K_1, K_2, \dots$). A user specializing in subset $K_m$ has a gain matrix $R$ with $R_{ii}=1$, $R_{ij}=0.2$ for all $j \in K_m$ ($i \neq j$), and all other off-diagonal entries being zero. The final utility class is formed by sampling a mix of these different specialists.
>
> We run our iterative patching/auditing style algorithm targeting each class. In addition, we are investigating the effect (if any) that other generic post-hoc calibration in the literature have on utility calibration in this case. We are currently conducting the experiments and expect to have the results by Friday 1st of August. Unfortunately, it is no longer possible to communicate plots, but we will be able to provide some of the results in table format (upon request, as we may not be able to post without answering a question).
>
>
> **Response to Q2: Comparative Discussion with Recent Literature**
>
>
> We thank the reviewer for pointing out several highly relevant papers. We agree that a detailed comparison is necessary to properly position our work. We will add a detailed discussion to our related work section.
>
>
> *   **Comparison with Noarov et al. (2023):** This work provides an online algorithm for the *online adversarial setting*, guaranteeing low regret for downstream agents who best-respond to its predictions. Its technical core is a bound on the cumulative *signed error* of its vector predictions, conditioned on events defined by the agent's *best-response action regions*. This guarantee is engineered to ensure the agent's policy achieves low *regret* over a sequence of interactions. In contrast, our work is designed for the *batch evaluation* of a given predictor and is agnostic to the agent's specific behavioral model. Our metric, ${UC}(f, u)$, is not a regret bound, guaranteeing the agent good utilities. Consequently, our guarantees are about the properties of the utility estimation rather than the agent policy.
>     Both works start from the same principle—that predictions should be unbiased conditional on events determined by downstream utilities—but they calibrate *different objects* and thus yield different guarantees, consequences and applications. In Noarov et al., event-conditional unbiasedness is imposed *coordinate-wise* on the whole prediction vector $p_t$: $|\sum_t (p_{t,i}-y_{t,i})E(x_t,p_t)| \le \alpha(n_t(E))$ must hold for every coordinate $i$ and every event $E$. Our paper instead collapses $p_t$ to a task-specific scalar $v_u(X)$ and requires that its worst-case interval bias $\sup_{I\subseteq[-1,1]}|\mathbb E[(u(f(X),Y)-v_u(X))\mathbf{1}\{v_u(X)\in I\}]|$ be small. This difference can cause both settings to widely deviate. For example, in our class-wise setting, Noarov et al. formulation would force the *entire* averaged prediction vector to match the averaged outcomes.
>
>
> *   **Comparison with Roth and Shi (2024):** In a similar direction, this work also focuses on the online adversarial setting, designing a forecasting algorithm to ensure low **swap regret** for all downstream decision-making agents. Their primary objective is to achieve superior *regret rates*. To do this, they design predictions that are unbiased only on a chosen set of events, and their guarantees apply specifically to agents who adopt certain behavioral models (e.g., (smooth) best-response). To this end, for dimension $1$ and $2$, they relax the requirement that the utility functions be known in advance, only requiring them to be $L$ Lipschitz. For higher dimensions, they require the best-responding agents to have known number actions and to *smoothly* best-respond.
>
>
> *   **Comparison with Hu and Wu (2024):**  This work focuses online (discrete/binned) binary prediction setting, proposing an efficient algorithm to minimize the *Calibration Decision Loss (CDL)*, which is equivalent to worst-case swap regret of best-responding agents. Their primary contribution is an online algorithm with provably near-optimal error rates for CDL. Most interestingly, their work shows a separation in regret rates between CDL and the Expected Calibration Error, i.e., it is possible to achieve a rate for CDL that breaks the lower bound on ECE. In contrast, our framework aims at achieving a scalable assessment methodology for multiclass models.
>
>
> *   **Comparison with Kimin et al (2023):** In the binary setting, U-Calibration framework is designed for *online, adversarial decision-making* with the goal of providing an online forecasting algorithm that guarantees low *external regret* for *any* rational agent. Technically, its error is the maximum possible regret over the entire class of *bounded proper scoring rules*. While we clearly align with some of the methodological choice of U-cal, our work differs in setting, approach, and the decision-theoretic consequence implied.
>
> Again, we are grateful for the reviewer suggestion to compare with this body of work  and we will add a version of this detailed discussion to the related work section to clearly situate our contribution.
>
>
> **Response to Q3, Presentation, Focus, and Novelty**
>
> We appreciate the reviewer's feedback on this. We will revise the manuscript to improve its clarity and better articulate our contributions.
>
> *   **Focus on Assessment:** Indeed our primary goal is to propose a scalable framework for **assessing** multiclass calibration. We will revise the introduction and section transitions to make this narrative clearer. The decision-theoretic benefits (Sec 3.1) are included to demonstrate *why* our metric is meaningful, and the post-hoc methods are included to show that achieving good performance on our metric is *possible*. We will modify our manuscript to keep the assessment narrative more central while *defining* the metric, *justifying* its value, and *showing* it is achievable. We will edit the text to make this flow more explicit for the reader. To further clarify our focus, we will revise the title to "Scalable Evaluation of Utility-Aware Multiclass Calibration."
>
> *   **Technical and Conceptual Novelty:**  We believe our core novelty lies in the synthesis and extension of existing tools to create a unified and scalable evaluation framework that was previously missing for multiclass models. While existing methods are often either computationally intractable in high dimensions or suffer from high bias, our framework bridges this critical gap. The concepts of "proactive" and "interactive" measurability are crucial for explaining why this gap exists and how our eCDF-based evaluation approach offers a practical and scalable solution. We will revise the manuscript to better highlight how this synthesis provides a valuable new perspective on the problem of multiclass calibration assessment.
>
> Once again, we thank the reviewer for their constructive and insightful feedback. We are confident that these revisions will substantially improve the paper, and we are eager to answer any follow-up questions during the discussion period.

---

> > ### Author Response · Authors · 2025-08-05
> >
> > Following up on our rebuttal, we have now completed the experiments on the sensitivity of our on aligned and misaligned utility class structures. For space constraints, we give a subsets of the results here for Algorithm 1 on ResNet and CIFAR100 against both "aligned" and "misaligned" utility classes (each constructed with 20 utility functions).
> >
> > The results are summarized below, where max error refers to the utility calibration error of the worst utility function and mean to the average.
> >
> > **Table 1: Convergence Speed of Algorithm 1**
> > | Iteration | Max Error (Aligned) | Mean Error (Aligned) | Max Error (Misaligned) | Mean Error (Misaligned) |
> > |:----------|:--------------------|:---------------------|:-----------------------|:------------------------|
> > | 0         | 0.0947              | 0.0930               | 0.0980                 | 0.0957                  |
> > | 50        | 0.0160              | 0.0144               | 0.0147                 | 0.0135                  |
> > | 125       | 0.0080              | 0.0070               | 0.0100                 | 0.0088                  |
> >
> >
> >
> > We also give the results for some standard post-hoc methods for comparison.
> >
> >
> > **Table 2: Final Error  Comparison**
> > | Method                | Max Error (Aligned) | Mean Error (Aligned) | Max Error (Misaligned) | Mean Error (Misaligned) |
> > |:----------------------|:--------------------|:---------------------|:-----------------------|:------------------------|
> > | Algorithm 1     | 0.0080          | 0.0070           | 0.0100             | 0.0088              |
> > | Uncalibrated    | 0.0947              | 0.0930               | 0.0980                 | 0.0957                  |
> > | Temperature Scaling   | 0.0111              | 0.0098               | 0.0115                 | 0.0097                  |
> > | Vector Scaling        | 0.0346              | 0.0331               | 0.0366                 | 0.0333                  |
> > | Matrix Scaling        | 0.0562           | 0.0547             | 0.0587              | 0.0563|
> >
> >
> > As shown, Algorithm 1 converges effectively for both utility structures and is more effective than standard calibration methods.   We will add the detailed results and visualizations to the revised manuscript.
> >
> > We are available to answer further questions regarding additional results or different experimental setting.

---

> > ### Comment · Reviewer_iTGh · 2025-08-05
> >
> > Thank you for your detailed response, and for including the new evaluation results. The tables give a high-level overview of these which I'm content with. I would have expected intuitively slower calibration error convergence on misaligned utilities whereas in fact both aligned and misaligned utilities resulted in similar error numbers with a few exceptions. That being said, the results still demonstrate in both cases reasonable trends in the calibration error, which further supports the paper's empirical side. I also appreciate the discussion of related works in the area of multiclass calibration that were pointed out in the review; while the motivation is similar, e.g. in the intention to reduce dimensionality based on the utility function-induced needs, several aspects indeed differ as explained.
> >
> > Furthermore, I appreciate the authors' ideas and overall willingness to refocus the presentation around the evaluation aspects of calibration. The proposed new title indeed sounds like a better description of the contribution, and I support it.
> >
> > For the future revision I would recommend further synthetic experiments on different utility classes (e.g. with different gain matrices) --- I agree with one of the other reviewers that the paper would have been further improved by providing truly practical experiments, but I still think that at least, more synthetic experiments will be valuable for breadth's sake. (Due to the short time available during the rebuttal period, I won't request these experiments now.)
> >
> > With my questions addressed, and in view of the proposed refocused improvements to the paper's presentation, I am happy to maintain my overall assessment of the paper.

---

### Official Review · Reviewer_GhnP · 2025-07-04

**Clarity:** 2
**Significance:** 3
**Originality:** 3
**Rating:** 5
**Confidence:** 3

**Summary:**

This paper proposes "utility calibration", which is a novel notion of calibration in classification that utilizes utility functions in assessment. The class of utility functions considered encapsulates the action rule by only taking in the predicted class probabilities and outcome realization as inputs, and it is assumed to be bounded. Utility calibration requires that the worst-case error in the expected utility be bounded, where the worst-case is determined by binning of the expected utility. The authors show that for a fixed utility function, computing utility calibration error is independent of the number of classes, and that it is achievable, and that it is polynomial in the number of classes when considering a class of utility functions. Empirical evaluations show that utility calibration provides another aspect to evaluating the goodness of distributional predictions.

**Questions:**

- Questions were included in "Weaknesses" section above.

**Ethical Concerns:**

["NO or VERY MINOR ethics concerns only"]

**Final Justification:**

Connecting uncertainty and calibration to its utility is an important topic, and I believe this work makes progress in this regard.

The discussion period helped to clarify any concerns I had about this work, and I support acceptance of this paper.

**Limitations:**

- Limitations are not discussed in the paper.

**Paper Formatting Concerns:**

- No major issues

**Quality:**

3

**Strengths And Weaknesses:**

Strengths
- Novel concept of calibration which considers how the model can be used downstream w.r.t a specific or class of utility functions
- Considering how useful a calibrated prediction is is a pertinent problem that many other works in uncertainty quantification or probabilistic forecasting does not touch upon, and this work provides a theoretical framework to connect the metric to the usefulness
- Proposed notion of calibration generalizes existing notions of calibration, while eliminating estimation issues with binning and dimensionality

Weaknesses
- It's unclear to me how different this notion is from decision calibration [14 (Zhao et al., 2021)]. Is it that decision calibration still depends on the number actions K?
- Why was decision calibration [14] left out from the empirical evaluations?
- Despite the theoretically well-motivated discussion on the notion and metric, the empirical evaluations seem to fall short in convincingly demonstrating the advantages provided by this metric.
  - In Table 1, is it possible at all to deduce that the utility-based versions of CWE and TCE are superior compared to the binned versions?
  - It would have been helpful to see the application of the models in a decision-making setting where the utility-based metrics actually provide a signal for higher utility of a model
  - I don't think any of the results actually demonstrate that the utility based metrics are "application-centric".
- Is it true that Lemma 3.4 only states the existence of such a recalibration algorithm, but the paper does not propose any algorithm that optimizes for utility calibration?

Minor points:
- L21: why the inequality for $\sum_i x_i$ instead of equality?
- L61-63: run on sentence
- L230: typo "out"
- L231: typo "classe"
- L308: typo "duch"
- Lines are very dense on page 7

---

> ### Author Rebuttal · Authors · 2025-07-30
>
> We thank the reviewer for their detailed and constructive feedback, which will help us significantly improve the clarity and impact of our work. We address each of the reviewer's points below.
>
> **Response to W1 & W2: Relationship to Decision Calibration [14]**
>
> We thank the reviewer for this important question. Our framework is indeed related to Decision Calibration (DC) [14], but differs conceptually and is designed to overcome its practical limitations.
>
> 1.  **Conceptual Difference:** DC explicitly models a scenario where a user chooses from one of $K$ actions. Its guarantee is that for any utility function over outcomes and $K$ actions, the estimated loss of an *optimal/best-response policy* derived from the model (and some utility function) is unbiased. In contrast, our framework abstracts away the explicit action space. The utility function $u(f(X), Y)$ is designed to encapsulate the entire decision-making procedure and its resulting utility. Instead of providing unbiasedness estimation guarantees for a set of policies, we provide a guarantee on the *predicted utility value itself*, ensuring that it is calibrated against the realized utility, particularly for the worst-case range of predicted utility values.
>
> 2.  **Practicality and Empirical Comparison:** We opted not to include DC in our empirical evaluation for two key reasons. First, as we cite from [11] Gopalan et al. (2024), formally measuring DC error is computationally intractable, with complexity scaling exponentially in the number of classes $C$, even for $K=2$. Second, while Zhao et al. (2021) propose a more tractable relaxation (Sec 4.4 in their paper), it involves solving a non-trivial optimization problem and comes with no formal guarantees that its solution corresponds to the true DC error. Given the lack of guarantees on the meaningfulness of this proxy, we opted not to compare against it. Our framework of utility calibration is motivated by these very challenges; we focus on *interactively measurable* utility classes to ensure our metrics are scalable and provide reliable assessments. We will clarify these points in the revised manuscript.
>
> **Response to W3, in particular W3.b & W3.c Empirical evidence and "Application-Centric" Nature**
>
> We thank the reviewer for their question, which touches on an important nuance of our work. We would like to clarify that *utility calibration does not guarantee a model will achieve higher utility.* Instead, it guarantees that a user's *assessment* of their expected utility is reliable and not systematically biased.
>
> This is very much in the spirit of calibration itself. Standard calibration ensures that a model's stated probabilities are reflective of real frequencies, but it does not guarantee high accuracy. Similarly, utility calibration ensures that a model's *predicted expected utility* is trustworthy, but does not guarantee the utility itself will be high. A poorly performing classifier can be utility-calibrated if its low expected utility predictions accurately reflect the low utility that is actually realized.
>
> The key benefit is preventing a user from being misled by overly optimistic (or pessimistic) forecasts about their strategy's performance. This provides a "what you see is what you get" guarantee. This reliability is the core of our "application-centric" claim.
>
> Further, it is useful to clarify what we mean by **application centric**: indeed, we do not directly incorporate real-life applications, (that would require domain experts to choose the utility functions). But we show how our approach naturally incorporates any choice of utilities, and detail several such choices in Sec 3.3.  Therein, we briefly discuss how some utility class can be relevant for particular domains. Figure 1 exactly illustrates our purpose: depending on the application and their choices, the users may have different utility classes, e.g. rank-based or linear. So depending on the application, user will be interested in the utility calibration given by Fig1a or by 1b (resp. 1c or 1d for ImageNet). Our eCDF plots then empirically demonstrate this by showing how this reliability holds or breaks down across various methods (Dirichlet, TS, etc). We will add this important clarification to Section 4.
>
>
> > However, as we noticed that reviewer GWKc was also puzzled by the use of "application centric" - we can rephrase as "application-dependent'', or more systematically `"utility-aware'' (as in the title) if the reviewer feels it makes the contribution clearer.
>
>
> **Response to W3.a: Superiority of Utility-Based CWE and TCE**
>
> We agree with the reviewer that one cannot deduce superiority from the numerical values in Table 1 alone. The superiority of our ${UC_{TCE}}$ and $UC_{CWE}$ metrics is *methodological and theoretical*. Binned estimators like $\mathrm{TCE}^{\mathrm{bin}}$ are known to be unstable and sensitive to heuristic choices, particularly the number of bins (e.g. Nixon et al. [20]), with different choices leading to conflicting conclusions. Our approach is binning-free; by taking the supremum over all possible intervals, we provide a single, worst-case error value that is robust to such arbitrary choices.
>
> To make this point more concrete, we commit to adding a new experiment to the appendix. This experiment will illustrate how the reported  binned estimators for a given model varies with the number of bins and binning strategy, in contrast to our stable, binning-free metric.
>
> **Response to W4: Recalibration Algorithm**
>
> We thank the reviewer for this question.  A key advantage of our framework is its direct compatibility with established, provably convergent algorithms. By framing utility calibration as a form of weighted calibration, our framework allows the adaptation of existing algorithms to improve a model's utility calibration. We provide a concrete example of this procedure in Appendix A (Algorithm 1) and will emphasize this practical aspect more prominently in main body.
>
> **Response to Minor Points**: we thank the reviewer for his feedback, that we will carefully take into account.
>
> **Limitations**
>
> The reviewer rightly points out the absence of a specific limitations section. **If our paper is accepted, we will add a dedicated section in the final manuscript.** We plan to use the additional page granted for the camera-ready version for this purpose. This section will discuss the computational challenges of *proactive measurability* for rich utility classes, our reliance on sampling for the eCDF evaluation, and directions for future work.
>
>
> Again, we thank you for your feedback. We remain available to answer additional questions if needed.

---

> > ### Author Response · Authors · 2025-08-02
> >
> > Following up on our rebuttal, we have now completed the experiments on the sensitivity of our on aligned and misaligned utility class structures. For space constraints, we give a subsets of the results here for Algorithm 1 on ResNet and CIFAR100 against both "aligned" and "misaligned" utility classes (each constructed with 20 utility functions).
> >
> > The results are summarized below, where max error refers to the utility calibration error of the worst utility function and mean to the average.
> >
> > **Table 1: Convergence Speed of Algorithm 1**
> > | Iteration | Max Error (Aligned) | Mean Error (Aligned) | Max Error (Misaligned) | Mean Error (Misaligned) |
> > |:----------|:--------------------|:---------------------|:-----------------------|:------------------------|
> > | 0         | 0.0947              | 0.0930               | 0.0980                 | 0.0957                  |
> > | 50        | 0.0160              | 0.0144               | 0.0147                 | 0.0135                  |
> > | 125       | 0.0080              | 0.0070               | 0.0100                 | 0.0088                  |
> >
> >
> >
> > We also give the results for some standard post-hoc methods for comparison.
> >
> >
> > **Table 2: Final Error  Comparison**
> > | Method                | Max Error (Aligned) | Mean Error (Aligned) | Max Error (Misaligned) | Mean Error (Misaligned) |
> > |:----------------------|:--------------------|:---------------------|:-----------------------|:------------------------|
> > | Algorithm 1     | 0.0080          | 0.0070           | 0.0100             | 0.0088              |
> > | Uncalibrated    | 0.0947              | 0.0930               | 0.0980                 | 0.0957                  |
> > | Temperature Scaling   | 0.0111              | 0.0098               | 0.0115                 | 0.0097                  |
> > | Vector Scaling        | 0.0346              | 0.0331               | 0.0366                 | 0.0333                  |
> > | Matrix Scaling        | 0.0562           | 0.0547             | 0.0587              | 0.0563|
> >
> >
> > As shown, Algorithm 1 converges effectively for both utility structures and is more effective than standard calibration methods.   We will add the detailed results and visualizations to the revised manuscript.
> >
> > We are available to answer further questions regarding additional results or different experimental setting.

---

> > ### Comment · Reviewer_GhnP · 2025-08-05
> > **Re: Rebuttal**
> >
> > Thank you, authors, for your detailed response. It was very helpful in clarifying many of my questions. I think the authors could still perform a comparison with decision calibration regardless of the lack of theoretical guarantees - I don't think that prevents experimentation and is of interest to the reader considering the relationship with the current proposed method. If the discrepancy with the current method is stark, that could be an interesting comparison point.
> >
> > Besides this point, I just wish there could be more emphasis on the practical aspects of the proposed framework. The main takeaways seem to be largely theoretical (which is obviously still very valuable), but the fact of considering utility (which is included in the title of the paper as well) brings about the expectation that there is a very practical application of the method proposed.
> >
> > I have no other concerns with this work.

---

### Note · Authors · 2025-08-16

We appreciate the reviewers’ careful and constructive engagement during the discussion; all reviewers maintained positive assessments of the paper. Our main contribution is a utility-calibration framework that evaluates reliability directly in terms of downstream utility, unifies and refines top-class and class-wise notions, extends calibration assessment to richer classes of downstream utilities, and provides scalable, binning-free metrics with strong guarantees.

Among the points clarified were the conceptual distinction from decision calibration, the framework’s application-dependent scope, and related sections such as the mechanics of Top-K versus linear/rank utilities. We have already acted on the feedback by expanding the related-work comparison and completing new experiments on aligned and misaligned utility classes.

Building on this, we will incorporate an additional small set of focused additions: a text-classification study to broaden empirical scope, a brief analysis of estimator stability versus sample size, and an empirical comparison with a tractable proxy for decision calibration  (while noting the formal hardness of measuring decision calibration and the limitations of such proxies). Alongside narrative and structural refinements, these updates will further strengthen the clarity and impact of the final manuscript.

Again, we are extremely grateful for the reviewers’ feedback and guidance.

---

### Decision · Program_Chairs · 2025-09-17

**Decision:**

Reject

**Comment:**

The paper proposes a new method for assessing the calibration of models by looking at the utility calibration of downstream applications. The authors show a scalable way to measure this calibration and how it generalizes commonly used calibration measures.

After discussions with the authors, all reviewers are in favor of accepting the paper. They appreciate the scalability of the proposed technique and its theoretical grounding. On the flip side, the reviewers felt that the novelty of the work is limited given recent work in decision calibration these past few years.